# Evaluating lubricant performance to reduce COVID-19 PPE-related skin injury

**Marc A. Masen**[1]*, **Aaron Chung**[1], **Joanna U. Dawczyk**[1], **Zach Dunning**[2], **Lydia Edwards**[1], **Christopher Guyott**[1], **Thomas A. G. Hall**[1], **Rachel C. Januszewski**[1], **Shaoli Jiang**[1,3], **Rikeen D. Jobanputra**[1], **Kabelan J. Karunaseelan**[1], **Nikolaos Kalogeropoulos**[1], **Maria R. Lima**[1], **C. Sebastian Mancero Castillo**[1], **Idris K. Mohammed**[1], **Manoj Murali**[1], **Filip P. Paszkiewicz**[1], **Magdalena Plotczyk**[4], **Catalin I. Pruncu**[1], **Euan Rodgers**[1], **Felix Russell**[1], **Richard Silversides**[1], **Jennifer C. Stoddart**[1], **Zhengchu Tan**[1], **David Uribe**[1], **Kian K. Yap**[1], **Xue Zhou**[1,5], **Ravi Vaidyanathan**[1]

**1** Department of Mechanical Engineering, Imperial College London, London, United Kingdom, **2** Department of Manufacturing Engineering, Coventry University, Coventry, United Kingdom, **3** Wuhan University of Technology, Wuhan, China, **4** Department of Bioengineering, Imperial College London, London, United Kingdom, **5** Southwest Jiaotong University, Chengdu, China

* m.masen@imperial.ac.uk

**Data Availability Statement:** The measurement data files are available from the Imperial College Institutional Research Data Repository DOI:10.14469/hpc/7269

## Abstract

### Background

Healthcare workers around the world are experiencing skin injury due to the extended use of personal protective equipment (PPE) during the COVID-19 pandemic. These injuries are the result of high shear stresses acting on the skin, caused by friction with the PPE. This study aims to provide a practical lubricating solution for frontline medical staff working a 4+ hours shift wearing PPE.

### Methods

A literature review into skin friction and skin lubrication was conducted to identify products and substances that can reduce friction. We evaluated the lubricating performance of commercially available products in vivo using a custom-built tribometer.

### Findings

Most lubricants provide a strong initial friction reduction, but only few products provide lubrication that lasts for four hours. The response of skin to friction is a complex interplay between the lubricating properties and durability of the film deposited on the surface and the response of skin to the lubricating substance, which include epidermal absorption, occlusion, and water retention.

### Interpretation

Talcum powder, a petrolatum-lanolin mixture, and a coconut oil-cocoa butter-beeswax mixture showed excellent long-lasting low friction. Moisturising the skin results in excessive friction, and the use of products that are aimed at '*moisturising without leaving a non-greasy*

**Funding:** The authors thank the Imperial College COVID-19 response fund for funding this study.

**Competing interests:** The authors declare no conflict of interest. All substances tested were obtained through normal commercial channels, except for some dressings, where the supplier provided a free trial version on their website, and one of the waxes, which was only available in the USA and was delivered to the investigators upon request. The investigators paid for the product, shipping and handling. The corresponding author had full access to data in the study and had final responsibility for the decision to submit for publication. This does not alter our adherence to PLOS ONE policies on sharing data and materials.

*feel* should be prevented. Most investigated dressings also demonstrate excellent performance.

## Background

### Covid-19, PPE and skin injury

Personal Protective Equipment (PPE) such as goggles, visors and respiratory protective equipment are a critical component in the defence against infectious diseases for frontline medical personnel. During the COVID-19 pandemic, the use of facial PPE is essential to limit the infection rate as the SARS-CoV-2 virus enters and infects human cells through respiratory, oral and ocular mucosal membranes via airborne transmission of contaminated droplets [1, 2]. This is particularly important for medical staff, who must be adequately protected during extended shifts. Refreshment of PPE poses increased virus exposure risk and as such, medical staff wear their facial PPE for much longer than recommended. The recent global COVID-19 pandemic has highlighted that prolonged use of facial PPE can cause a range of skin issues, including irritation and injuries such as skin tears, pressure injuries and urticaria [3]. Maintaining skin health and preventing injury whilst using PPE has become an important aspect of advice to medical staff [4]. Preventing skin injury has become increasingly critical as recent findings show that there also is a dermal pathway to COVID-19 infection [5].

In a survey by Jiang et al. on skin injuries incurred by medical staff during the COVID-19 pandemic, respondents included the nasion, cheek bones and the forehead amongst the main affected anatomical sites [6]. Elevated shear strains are observed at bony prominences, explaining why PPE-induced facial skin injury is often found at these sites [7]. They concluded that heavy sweating, increased duration of PPE use, the application of higher-grade PPE, and being of the male gender were significant risk indicators for skin injury to occur. Heavy and airtight PPE inhibits perspiration to volatilise, thus changing the microclimate and decreasing the tolerance of skin to injury [8, 9]. Foo et al. reported high rates of detrimental skin reactions amongst N95 face mask users and concluded a need to investigate alternate means to prevent injury [10].

### Relating the clinical issues to biomechanics

The negative effects of shear forces acting on the skin have long been established, Reichel demonstrated in 1958 that skin injury develops at lower pressures when a shear force is applied to the skin [11]. In addition, excessive shear can affect the integrity of skin and the combination of high shear stresses and weakened structure of the skin will result in substantial tissue deformation and cellular distortion [12]. Shear loading decreases blood perfusion and transcutaneous oxygen level in skin, making it more prone to injury [13]. These findings, as schematically summarised in Fig 1B, mean that controlling the level of shear in the skin-PPE interface is critical to the prevention of injury. Shear stresses occur at the skin-PPE interface due to three primary mechanisms: (i) expansion of the PPE material in the direction perpendicular to loading; (ii) sticking of the contacting bodies due to static friction that prevents initial sliding, and to a lesser extent (iii) dynamic friction due to macroscopic sliding of the contacting bodies. These three mechanisms may occur due to motion of facial features and adjustments of the PPE. This is also recognised in recent literature and advice to medical staff, which include half-hourly application of a lubricant or a moisturiser to the skin [4, 14].

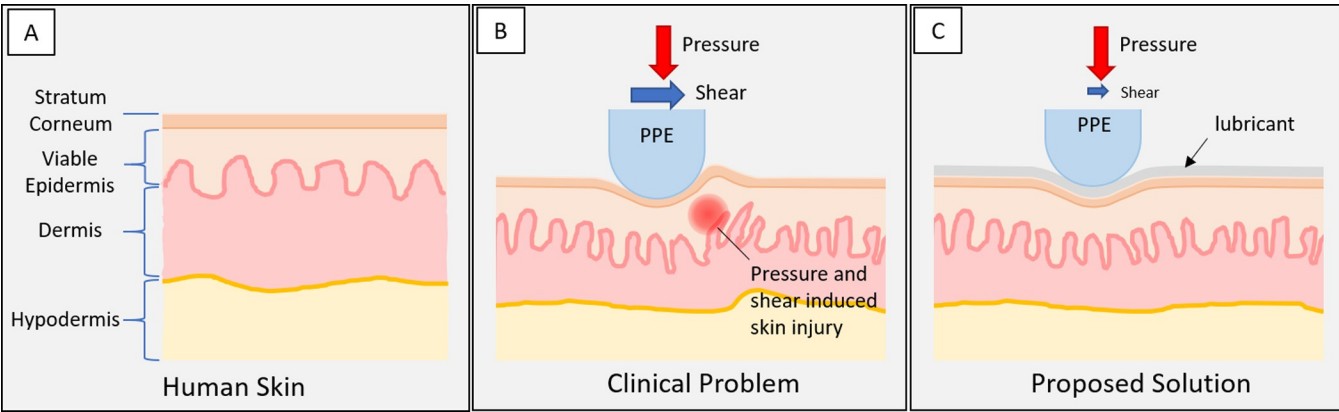

**Fig 1. The proposed solution of a non-absorbent, non-occlusive lubricant film at the PPE-skin interface, resulting in a lower risk of damage when wearing PPE for extended periods of time.** A: Schematic overview of the three main layers of human skin. B: The clinical problem of skin injury occurs because of elevated mechanical shear stresses acting on the skin. C: Skin lubrication leads to reduced friction and lower mechanical stress in the skin.

### A biomechanics-based solution to shear-induced skin injury

It can be concluded that understanding the interaction between skin and PPE is essential to prevent injury [15]. In this work we present our investigation into the shear loads caused by static friction in a skin-polydimethylsiloxane (PDMS) interface under conditions replicating that of a PPE-skin contact. PDMS was chosen as it is commonly used in PPE as the compliant material that interfaces with skin. The objective is to provide hospital staff with advice and practical solutions when they are required to wear facial PPE for prolonged periods of time. This study provides evidence that the application of a commercially available, inexpensive, and dermatologically safe lubricant at the skin-PPE interface results in lower shear stresses over extended periods of time and could therefore relieve skin injuries caused by PPE (Fig 1C).

## Methods and materials

The study was approved by the Imperial College London Science, Engineering and Technology Research Ethics Committee (SETREC), reference 20IC5999.

### Tribometer

A portable tribometer with a wheel-shaped probe is used in this work (Fig 2A) [16]. The motorised wheel is placed against the skin, creating a shear force upon rotation. The motor is connected to two perpendicularly mounted force transducers that measure the applied normal load and the resulting shear forces in the contact. The system is free to move vertically through sliding bearings whilst horizontal motion is restricted. The load on the skin-probe contact is applied through dead-weight loading and, when the device is used upright, equals the combined weight of the probe, motor, force transducers and the slider. The probe has a width of 10 mm and a diameter of 33 mm. It comprises a 25 mm aluminium core, covered with a 2 mm adhesive tape and a 2 mm platinum cured PDMS layer (Silex, Hampshire, UK). The rotational velocity of the probe is 10·6 rpm, resulting in a sliding speed of 18·4 mm s$^{-1}$ at the skin interface.

Brill et al. state that contact pressures for ventilation masks range between 8 and 30 kPa [17]. Kuilenburg suggests an effective modulus of elasticity of 50 kPa for skin at this length scale [18]. A load of 1·1 N in combination with the dimensions and properties of the probe results in a mean contact pressure of 20 kPa. Experiments were performed on the skin of the

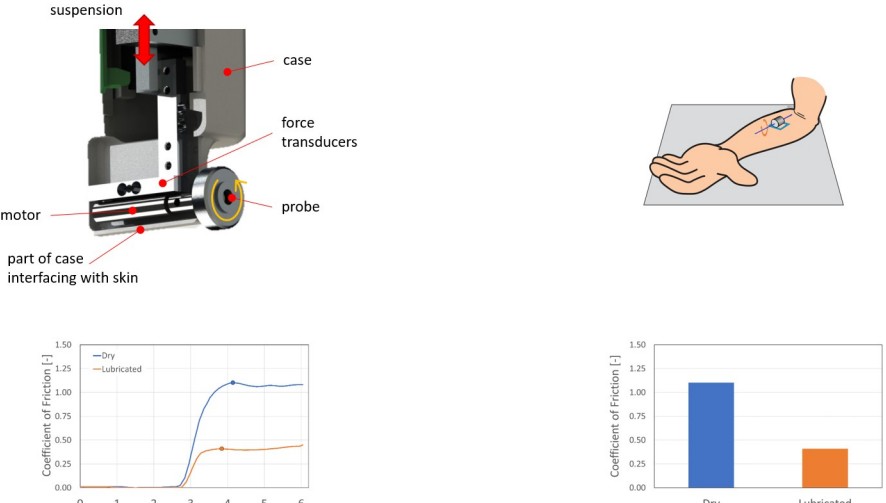

**Fig 2. Schematic overview of the measurement protocol.** A: Schematic of the interior of the tribometer setup. B: Schematic illustration of the measurement, adapted from Veijgen.[16] C: Two typical friction measurements. D: Final processed data.

left (non-dominant) arm, with the sliding direction of the wheel from the radial to the ulnar side. The normal and shear forces during the experiments were measured and a resulting coefficient of friction was calculated. Fig 2C shows two typical curves for a dry (blue) and a lubricated (orange) contact. The maximum coefficient of friction during the initial stage of sliding, representing the transition from static to dynamic friction, was reported (Fig 2D). Each experiment was performed three times, and in the results (Fig 3) the standard deviation is shown using error bars.

## Participants

Due to COVID-19 restrictions, performing in vivo tests was limited to a single participant. The participant, Caucasian male, healthy, age 44 years, mass 80 kg, height 1.84 m, gave informed consent. Our previous work has shown large interpersonal differences in friction for unlubricated skin, which are related to variations in sebum, hydration and mechanical properties [16, 19]. Although exact values may vary between people, Derler et al. demonstrated that the observed trends in friction are repeatable for variations of the lubrication and skin hydration [20]. Additionally, Veijgen et al. and Falloon et al. found that age was not a major parameter affecting friction [16, 21]. The experimental programme was performed over a one month period, from mid-May to mid-June 2020.

## Friction reducing agents

Criteria for products to be included in this investigation were (i) commercial availability, and (ii) safe for topical application. A longlist of commercially available creams, balms, powders, and ointments with a wide range of ingredients was composed, from which a shortlist was compiled based on direct availability and minimisation of duplication of main ingredients. The resulting shortlist of products was categorised into four groups depending on their appearance: "creams and grease-like lubricants", "wax-like lubricants", "powders", and "dressings and thin solid films". Table 1 lists the products used, what they are referred to in this manuscript and their ingredients.

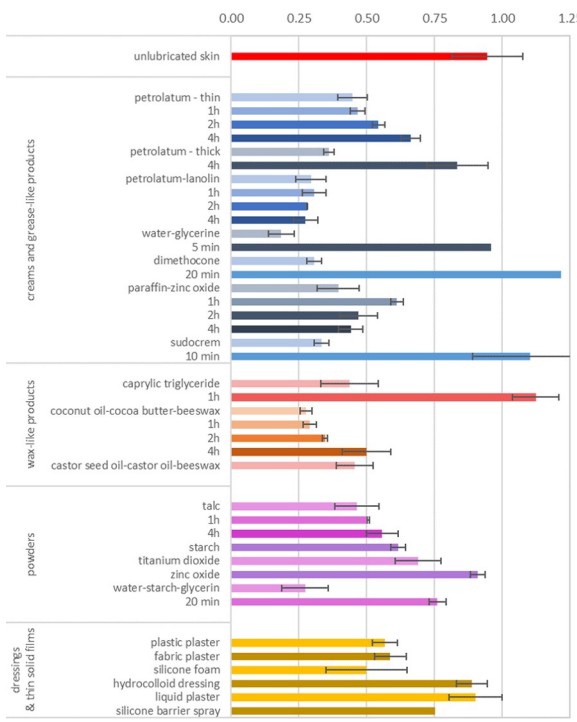

**Fig 3. Overview of results.**

## Protocol

Experiments were performed on the left (non-dominant) volar forearm, which is commonly used as a test site [19–23]. Hendriks et al. reported no significant difference in friction values measured on the forearm and the cheek for a range of materials and conditions [22]. The forearm was placed in a relaxed position onto a solid horizontal surface, schematically illustrated in Fig 2B. The casing of the tribometer was placed onto the forearm, also initiating contact between the probe and the skin test-site. After a waiting period of 2·5 s, rotation of the probe was initiated, creating a sliding interface between the probe and the skin surface.

48 hours before commencing the experimental programme, hair was removed from the skin by strip waxing. This procedure was repeated every five days to keep the skin clear of hair, after which no experiments were performed for 12 hours. At least one hour prior to an experiment, the skin was cleaned using water and a non-soap body wash (Aveeno Daily Moisturising, Johnson & Johnson, USA), rinsed and dried using a cloth towel. No further skin care regimen was followed. The substance under investigation was applied to the skin as specified in the product information leaflet that was supplied with most products. For most liquid substances, and for the cases where the amount to be applied was not specified clearly, this means that approximately 0·05 ml was applied at a location two-thirds of the distance from the wrist to the elbow. The substance was distributed and rubbed in using the index finger of the other hand over a 4 x 6 cm$^2$ area marked with a permanent marker, resulting in a coverage of approximately 2 mg/cm$^2$. For petrolatum to be effective, it has been suggested that a thick layer should be applied[14]. This advice was also followed by applying about 0·5 ml to the skin. Powders were deposited liberally onto the skin, rubbed in with the index finger of the other hand for 5 s and any excess powder was removed by briefly blowing onto the surface of the skin.

**Table 1. Summary of all lubricating agents tested.**

| | Lubricant | Ingredients as listed on packaging | Commercial name | Manufacturer |
|---|---|---|---|---|
| **Creams and grease-like lubricants** | Petrolatum | Petrolatum, bht, tocopheryl acetate | Vaseline | Unilever |
| | Petrolatum, lanolin | Petrolatum, lanolin, mineral oil, caprylyl glycol, glycine soja oil, parfum, ricinus communis seed oil, salicylic acid, tocopherol, zea mays oil, bht, citral, citronellol, geraniol, limonene, linalool, phenoxyethanol, iron oxides (ci 77491, ci 77492) | Eight Hour Cream | Elizabeth Arden |
| | Glycerine, water | Aqua, glycerine, propylene glycol, hydroxyethylcellulose, methylparaben, sodium phosphate, disodium phosphate, propylparaben, tetrasodium edta | K-Y Lubricating Jelly Sterile | Reckitt Benckiser |
| | Silicone | Dimethicone, dimethiconol | Silicone Lube | So Divine |
| | Paraffinum, zinc oxide | Paraffinum liquidum, zinc oxide, polyethylene, avena sativa kernel flour, sorbitan isostearate | Baby Daily Care Barrier Cream | Aveeno |
| | Zinc oxide, lanolin | Zinc oxide, benzyl alcohol, benzyl benzoate, benzyl cinnamate, lanolin, purified water, liquid paraffin, paraffin wax, beeswax, microcrystalline wax, sodium benzoate, linalyl acetate, propylene glycol, citric acid, butylated hydroxyanisole, sorbitan sesquioleate, lavender fragrance | Sudocrem | Forest Tosara |
| **Wax-like lubricants** | Coconut oil, cocoa butter, beeswax | Cocos nucifera oil, theobroma cacao seed butter, cera alba, tocopherol | Anti-Chafe Salve | Squirrel's Nut Butter |
| | Caprylic triglyceride, ozokerite wax | Caprylic/capric triglyceride, cetearyl acetate, ozokerite wax, glyceryl behenate, stearyl alcohol, allantoin, cocos nucifera, prunus dulcis oil, tocopherol, glyceryl linoleate & glyceryl linolenate | Face Glide | Body Glide |
| | Castor seed oil, beeswax, coconut oil, cocoa butter | Ricinus communis seed oil, hydrogenated castor oil, beeswax, cocos nucifera oil, peg/ppg-18/18 dimethicone, theobroma cacao butter, allantoin, citrus paradisi essential oil, citrus medica limonum peel essential oil, citrus aurantifolia essential oil, litsea cubeba fruit essential oil, citrus reticulata essential oil, citrus aurantium dulcis peel oil | Prosthetic Salve | Resilience |
| **Powders** | Talcum powder | Talc, parfum | Baby Powder | Johnson & Johnson |
| | Corn starch | Zea mays starch, gossypium herbaceum, hydroxyapatite, parfum | Cottontouch powder | Johnson & Johnson |
| | Zinc oxide | Zinc oxide | Zinc Oxide | BiOrigins |
| | Titanium dioxide | Titanium dioxide e171 | Icing Whitener | Sugarcraft Essentials |
| | Water, starch, glycerine | Water, potato starch, glycerin, stearic acid, cetyl alcohol, sunflower seed oil, sweet almond oil, propylene glycol, aminomethyl propanol, magnesium aluminum silicate, dimethicone, carbomer imidazolidinyl urea, methylparaben, propylparaben, aloe barbadensis leaf juice, sodium citrate, tocopheryl acetate | Liquid Powder | Resilience |
| **Dressings & thin solid** | Plastic dressing | Non-specified 'plastic' | ref 45906 | Elastoplast |
| | Fabric dressing | Non-specified 'textile' | ref 02607 | Elastoplast |
| | Silicone foam dressing | Polyurethane foam and polyurethane film | Silicone Adhesive Foam Lite | ActivHeal |
| | Hydrocolloid dressing | Thin polyurethane film | Comfeel Plus Transparent | Coloplast |
| | Germolene | Ethyl acetate, alcohol denat, nitrocellulose, ricinus communis, isopropyl alcohol, amyl acetate, isobutyl alcohol, camphor, parfum (includes benzyl alcohol, citronellol) | Liquid Plaster | Germolene |
| | Cavilon | Hexamethyldisiloxane, isooctane, acrylate terpolymer, polyphenylmethylsiloxane | Cavilon Barrier Film | 3M |

If a test provided a much-elevated friction compared to other products in the same category, then further tests using this product were not conducted. Additionally, if a test showed a strong increase of friction with time, no further measurements at other time-intervals were performed. After each experiment, the product was removed from the skin using a gentle wipe with a paper tissue and the skin was cleaned using water and non-soap body wash, followed by a recovery period of at least 3 h before another experiment was commenced. A friction measurements on untreated, or 'dry' skin was performed at the start of every test day and, although dry friction results are inherently somewhat more variable than lubricated experiments as

evidenced by the error bars in Fig 3, no deviating trends were observed. This indicates that there was no significant change in the friction behaviour of the skin during the course of the experimental programme.

**Lubricant performance over time.** In this contribution, we aim to provide a solution that will reduce PPE-skin shear forces over the duration of an extended shift of four hours. It is paramount to understand not only the instantaneous effectiveness but also the evolution over time. All lubricants listed in Table 1 were tested and, based on their friction performance directly upon application and after 5–10 minutes, a selection was made for testing over a four-hour period. In between tests the skin with the applied product was covered by a piece of PDMS which was held in place using four fabric plasters.

## Role of the funding source

The research funder did not play a role in the research.

## Results

Fig 3 shows the measured coefficient of friction for the various products, alongside the value for unlubricated skin in the topmost red coloured bar. Results are grouped by product category: 'creams and grease-like products', 'wax-like products', 'powders' and 'dressings and thin solid films'. The reported coefficient of friction for each product was measured directly upon application, and 1, 2 and 4 hours after application.

Most products provide a low friction directly upon application and a reduction in friction to 20% of the unlubricated situation is observed. For some products, this low initial friction is followed by a rapid increase over time, examples include 'silicone-based' and 'water and glycerine-based' products, for which the measured friction exceeds the original dry friction by 1% and 29%, respectively. Other products, such as petrolatum show a more gradual increase over time, providing an initial friction reduction to 47% of the dry value, which increases again towards to dry value after approximately four hours. Powders with a liquid carrier generally show low instantaneous friction of approximately 50% of the dry friction, followed by an increase over time once the carrier has evaporated. Most dry powder products did not provide sufficiently low values to warrant investigating their longer-term performance, except talcum powder which provides consistently low friction, rising from 49% of the dry friction value directly after application to 59% at 4 hours after application. The various dressings tested showed moderately reduced levels of friction, with levels at approximately 60% of the value for dry friction. Overall, persistent low friction was obtained for the duration of the entire four-hour test for talcum powder (49%– 59%), the petrolatum-lanolin mixture (30%) and the coconut oil-cocoa butter-beeswax mixture (31%– 53%).

## Discussion

These observations described above are in line with literature [19–23]. It is worth noting that friction is a system property that not only depends on the lubricant, but also on the materials in contact, the applied pressure, and the environmental conditions. PPE that interfaces with skin often uses a compliant "rubber-like" material such as PDMS or a thermoplastic elastomer to reduce discomfort due to normal loading. However, most published work into contact mechanics and friction investigates hard specimens such as steel, glass or engineering plastics and focuses on the sliding or dynamic friction [19–23]. Therefore, the applicability of existing tribological knowledge to PPE-skin interactions is limited.

In the presented results, long-lasting low friction was observed for talcum powder, the coconut oil-cocoa butter-beeswax mixture and the petrolatum-lanolin mixture, whilst a range of

products with almost similar composition demonstrated only brief-lasting low friction. The friction in the skin-PPE contact depends on the interplay between the lubricant, the skin and the PPE material and includes the persistence of the product in the interface as well as the adsorption into the stratum corneum. The total friction response is the combination of the forces related to viscoelastic deformation of the bulk material, $F_{bulk}$, and the forces related to breaking of intermolecular bonds on the interface between the two material, $F_{int}$, as shown in Eq (1). Adams et al. demonstrated that for contacts between skin and other surfaces the interfacial friction dominates the total frictional effects, whilst viscoelastic effects may be ignored: [23]

$$F_f = F_{bulk} + F_{ink} \approx F_{int} \tag{1}$$

The interfacial friction force, Eq (2), is the product of the area of contact $A$ between the two surfaces and the shear strength of the interface $\tau$:

$$F_{int} = \tau \cdot A \tag{2}$$

Whilst this appears to be a straightforward relationship, i.e. lubricants directly affect $\tau$ thus causing low friction, the complexity is that with time the lubrication effect may decrease, whilst adding a substance to the skin may also have a pronounced effect on the contact area through various mechanisms, as illustrated in Fig 4A–4F.

## Creams and grease-like lubricants

Compared to the unlubricated situation of Fig 4A, applying a cream or grease-like product to the skin results in an instantaneous low friction, because of the reduced interfacial shear strength, as illustrated in Fig 4B. A gradual increase in coefficient of friction is seen for petrolatum, whilst the silicone and glycerine-water demonstrate a rapid increase. The longer-term response depends on the interaction of the product and skin. Dimethicone and petrolatum are highly efficient skin occlusives that prevent transepidermal water loss (TEWL) within 30 minutes of application [24, 25]. The entrapped water reduces the stiffness of the skin, whilst epidermal uptake of petrolatum or dimethicone is minimal, resulting in the situation described in Fig 4C [26]. Petrolatum is quite persistent on the skin surface, represented by Fig 4C, whilst dimethicone is more easily rubbed off, resulting in Fig 4D. Comparing results obtained for petrolatum and the petrolatum-lanolin mixture suggests that the initial lubricating function of these two products is similar. Lanolin is an emollient that resembles natural skin lipids and is linked to lubricity [27]. The obtained results suggest that compared to neat petrolatum, the addition of lanolin provides more enduring lubrication, whilst is still being highly occlusive, as evidenced by the friction traces shown in Fig 4B. A third lubricant that shows constant results is the liquid paraffin and zinc oxide containing cream. Similar to the wax, described later in Fig 5C, it appears to be non-occlusive and forms a non-absorbing and lubricating white film on the surface, a situation illustrated by Fig 4B.

Humectants attract water to the stratum corneum and the epidermis from the dermis, softening the skin within 15 minutes upon application [28, 29]. Commercial moisturising creams are often designed to not leave a greasy residue. This leads to a situation which, in terms of reducing shear, is a highly undesirable combination of no lubrication, and thus a high value of $\tau$ and the skin being plasticised and soft, resulting in a large contact area $A$. This combination causes excessively high shear forces in the skin-PPE interface as schematically illustrated in Fig 4D. A typical friction profile measured for a moisturising cream (Nivea men creme, Beiersdorf, Germany) is shown in Fig 5A. Upon application friction is low, but 10 minutes after application a very high static friction peak can be observed.

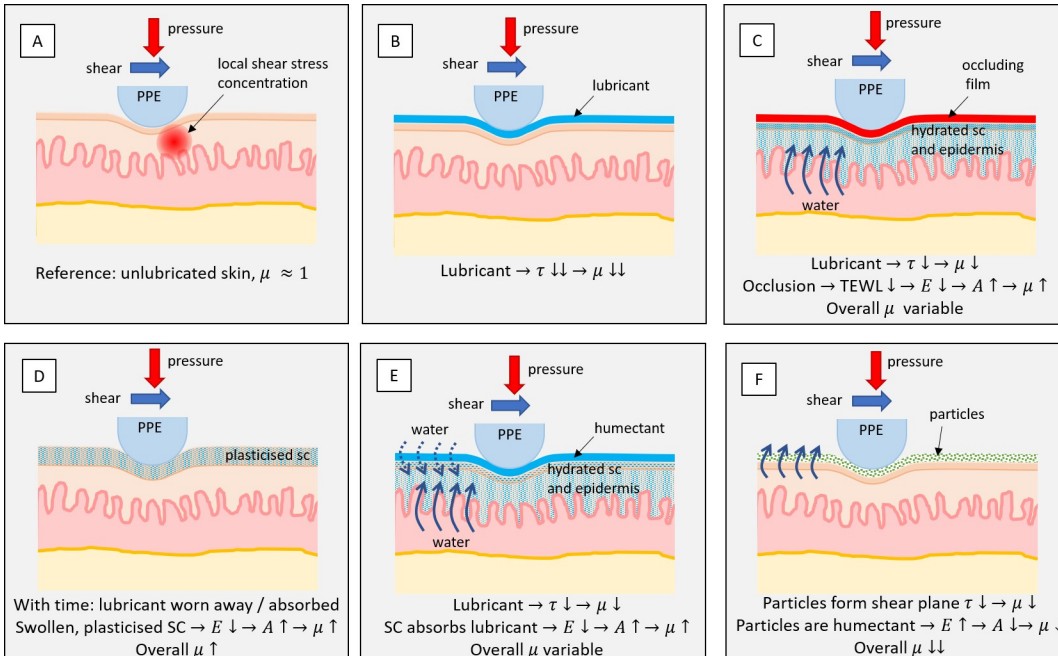

**Fig 4. Mechanisms involved in lubricating the skin.** A: In the unlubricated situation the friction mainly arises mainly from lipids on the skin surface. The exact value varies strongly between people, but in general the coefficient of friction is close to 1. B: Lubricants are highly effective at reducing the shear strength of the interface, resulting in much reduced friction. C: If the lubricating substance occludes the skin, transepidermal water loss is prevented. This hydrates the epidermis from the inside, reducing the stiffness and increasing the contact area thus increasing the coefficient of friction. D: The lubricant may not persist due to absorption, evaporation and/or wear. Lubricant absorbed into the stratum corneum still affects swelling and stiffness, causing the friction to increase. E: The lubricant may absorb into the skin, swelling and plasticising the stratum corneum (SC). This will reduce the stiffness and increase the contact area. The coefficient of friction will increase with time. F: Particulates may reduce the coefficient of friction: a lamellar structure provides low shear strength whilst round particles act as rollers. Some particles absorb moisture, increasing the stiffness of the stratum corneum. The combination of these effects may lead to a strong reduction of the coefficient of friction.

## Wax-like substances

Wax-like substances generally demonstrate low friction with excellent longevity. These products comprise triglycerides, fats and lipids. Low friction was observed for a mixture of coconut oil, cocoa seed butter and beeswax. Fig 5C shows friction traces before and 30 minutes after application of this wax, as well as two minutes after subsequent removal of the remaining wax

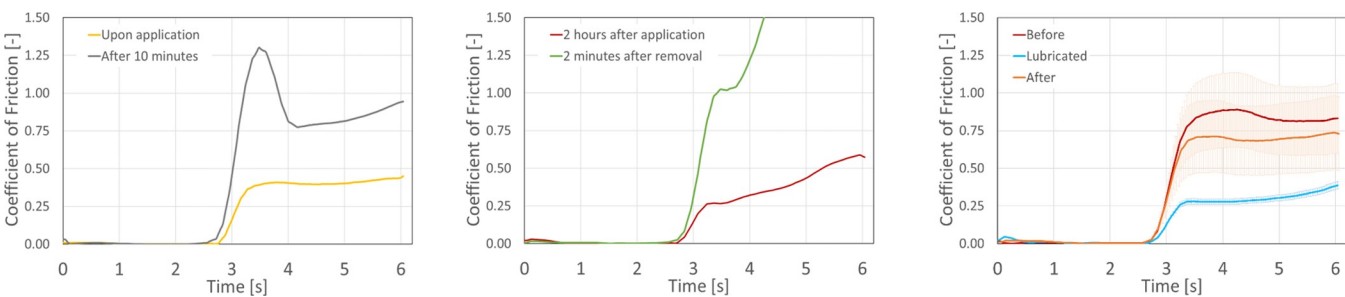

**Fig 5. Examples of various observed friction traces.** A: Friction trace observed for a moisturiser upon application and after 10 minutes: high static friction due to absorption into the skin. B: Friction trace for petrolatum+lanolin 2h after application, followed by removal using an alcohol wipe and a 5-minute wait, indicating occlusion. C: Friction trace before applying and after removal of wax, indicating the friction returns to the initial state and no occlusion occurs.

using an alcohol wipe. If occlusion had occurred, this would have resulted in a much-higher friction curve, compare e.g. with the green curve in Fig 5B. Because the friction response returns to its initial 'dry' value, it is concluded that occlusion did not occur and the mechanism behind the friction response is a durable low-shear layer (Fig 4B). It is noteworthy that the best performing wax appears to have a melting temperature just below skin temperature, meaning the deposited layer is in the liquid state.

## Powders

No obvious trends were found for the various powders, which were all selected for their moisture absorbent capabilities. Talcum powder showed low friction, in contrast to starch, zinc oxide and titanium dioxide. Therefore, the effect of absorption of moisture from the stratum corneum appears negligible. Deacon et al. related the low friction of talc to its lamellar structure, which is similar to solid lubricants like graphite and molybdenum disulphide [30]. The conclusion is that the friction behaviour for powder-lubricated skin-PPE contact is determined by the powder's capacity to reduce $\tau$ as illustrated in Fig 4F.

## Dressings & sprays

Most tested dressings demonstrated a relatively high friction against the PDMS probe. However, dressings provide protection either by functioning as a thick, compliant layer absorbing shear loading through deformation, or as a stiff layer distributing stresses over an extended area. Dressings may therefore be a simple and efficient method to reduce the shear loading on the skin. It needs noting that in that case any mask fit testing should be redone with the dressings in place. None of the tested spray-on films had sufficient thickness or stiffness to provide protection from shear forces and most tested products had a tacky appearance, resulting in a high friction situation resembling Fig 4E.

## Limitations of this research

The in vivo experimental programme was performed on a simulated PPE-skin tribo-system, using only one subject. Whilst quantitative results vary between people, literature shows that qualitative trends in friction are repeatable [16, 20, 21]. The mechanical aspects of the tested system match those in a skin-PPE interface, one silicone material was used to represent the wide variety of PPE available. The volar forearm was used as a surrogate for facial skin, and it should be noted that there are differences between the two sites in terms of topology, composition, and lipids. Perspiration was not taken into account as there is a large variation between people. Finally, care should be taken that the use of lubricant does not affect the functionality and sealing capacity of the PPE. The main contribution of the presented work is in the observed qualitative trends, which serve to inform on methods to reduce shear-induced injury by adapting the skin care regime.

## Conclusions

Reduced shear stresses acting on the skin are critical to ensure the comfortable and injury free use of PPE. A range of commercially available lubricating substances were investigated for their application as a shear reducing agent in skin-PPE contacts. Results indicate that the use of emollients and moisturising creams is to be discouraged when wearing PPE for long durations as they may result in excessive shear forces acting on the skin. Talcum powder, a lanolin containing petrolatum, and a coconut oil-cocoa butter-beeswax mixture provide excellent long-lasting lubrication.

## Supporting information

**S1 File.**
(PDF)

## Acknowledgments

The authors gratefully acknowledge support from the UK DRI-CRT and the Departments of Mechanical Engineering and Bioengineering at Imperial College London.

## Author Contributions

**Conceptualization:** Marc A. Masen, Aaron Chung, Joanna U. Dawczyk, Lydia Edwards, Christopher Guyott, Thomas A. G. Hall, Rachel C. Januszewski, Shaoli Jiang, Rikeen D. Jobanputra, Kabelan J. Karunaseelan, Maria R. Lima, C. Sebastian Mancero Castillo, Idris K. Mohammed, Manoj Murali, Filip P. Paszkiewicz, Magdalena Plotczyk, Catalin I. Pruncu, Euan Rodgers, Felix Russell, Richard Silversides, Jennifer C. Stoddart, Zhengchu Tan, David Uribe, Kian K. Yap, Xue Zhou, Ravi Vaidyanathan.

**Data curation:** Marc A. Masen.

**Formal analysis:** Marc A. Masen, Rikeen D. Jobanputra, Manoj Murali, Kian K. Yap, Xue Zhou.

**Funding acquisition:** Marc A. Masen.

**Investigation:** Marc A. Masen, Rikeen D. Jobanputra, Manoj Murali, Kian K. Yap, Xue Zhou.

**Methodology:** Marc A. Masen, Zach Dunning, Nikolaos Kalogeropoulos.

**Project administration:** Marc A. Masen.

**Resources:** Marc A. Masen, Zach Dunning, Nikolaos Kalogeropoulos.

**Software:** Marc A. Masen.

**Supervision:** Marc A. Masen.

**Validation:** Marc A. Masen.

**Visualization:** Marc A. Masen, Zhengchu Tan.

**Writing – original draft:** Marc A. Masen, Rikeen D. Jobanputra, Manoj Murali, Zhengchu Tan, Kian K. Yap, Xue Zhou.

**Writing – review & editing:** Marc A. Masen, Aaron Chung, Joanna U. Dawczyk, Zach Dunning, Lydia Edwards, Christopher Guyott, Thomas A. G. Hall, Rachel C. Januszewski, Shaoli Jiang, Rikeen D. Jobanputra, Kabelan J. Karunaseelan, Nikolaos Kalogeropoulos, Maria R. Lima, C. Sebastian Mancero Castillo, Idris K. Mohammed, Manoj Murali, Filip P. Paszkiewicz, Magdalena Plotczyk, Catalin I. Pruncu, Euan Rodgers, Felix Russell, Richard Silversides, Jennifer C. Stoddart, Zhengchu Tan, David Uribe, Kian K. Yap, Xue Zhou, Ravi Vaidyanathan.

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
