## [Decision Letter · Decision Letter 0]

13 Aug 2020

PONE-D-20-21415

Evaluating lubricant performance to reduce COVID-19 PPE-related skin injury

PLOS ONE

Dear Dr. Masen,

Thank you for submitting your manuscript to PLOS ONE. After careful consideration, we feel that it has merit but does not fully meet PLOS ONE’s publication criteria as it currently stands. Therefore, we invite you to submit a revised version of the manuscript that addresses the points raised during the review process.

ACADEMIC EDITOR:

I received comments and recommendations from three reviewers. Two reviewers are medicial experts and one is the expert in mechanics. Their recommendations are mixed but the comments may be helpful for further improvement.Your research is interesting and we hope the result of this research may be helpful to the current community. Therefore, please also pay some attentions to the potential by-effects. 

We look forward to receiving your revised manuscript.

Kind regards,

Jianguo Wang, PhD

Academic Editor

PLOS ONE

Reviewers' comments:

Reviewer's Responses to Questions

**Comments to the Author**

1. Is the manuscript technically sound, and do the data support the conclusions?

Reviewer #1: Yes

Reviewer #2: No

Reviewer #3: Yes

2. Has the statistical analysis been performed appropriately and rigorously? 

Reviewer #1: Yes

Reviewer #2: No

Reviewer #3: N/A

3. Have the authors made all data underlying the findings in their manuscript fully available?

Reviewer #1: Yes

Reviewer #2: No

Reviewer #3: Yes

4. Is the manuscript presented in an intelligible fashion and written in standard English?

Reviewer #1: Yes

Reviewer #2: Yes

Reviewer #3: Yes

5. Review Comments to the Author

Reviewer #1: Healthcare workers around the world are experiencing skin injury due to the extended use of personal protective equipment (PPE) during the COVID-19 pandemic. This study aims to provide a practical lubricating solution for frontline medical staff working a 4+ hours shift wearing PPE.

A literature review into skin friction and skin lubrication was conducted to identify products and substances that can reduce friction. We evaluated the lubricating performance of commercially available products in vivo using a custom-built tribometer. The results can improve our understanding of evaluating lubricant performance to reduce COVID-19 PPE-related skin injury. The detailed comments are presented below:

1. The study was approved by the Imperial College London Science, Engineering and Technology Research Ethics Committee (SETREC), reference 20IC5999. Please provide evidence.

2. In this study, the experimental programme was performed over a one month period, from mid-May to mid-June 2020. Please list your work for each period in detail

3. In this study, there are some formulas. But the formula is not numbered. Besides, each variable needs to be explained.

4. There are so many authors for this article. A statement for your authors' contribution and conflicts of interest are necessary.

Reviewer #2: This is a very interesting experiment.In this study, a 44 year old caucasian man was recruited to test the effectiveness of a series of lubricants to reduce friction and shear force in the left forearm (instead of facial skin), and hope find which lubricants has a protective effect for more than 4 hours.

However, the experimental design did not consider the following factors:

1.Participants: The subjects' health status was unknown, such as height, weight, etc.Only one caucasian male was selected and does reflect the friction and lubricant effect of women and other skin color health care workers on their skin ?

2.Setting: The volar forearm was selected as the experimental site. Does affect the friction coefficient measured after using the lubricant due to the different thickness of between the stratum corneum on the volar forearm and the facial skin ?Does affect the friction coefficient measured after lubricant use due to different perspiration of facial skin and forearm skin?In this study, authors measured the static friction coefficient , while the skin of medical staff wearing PPE against covid-19 produced dynamic friction coefficient. How to explain the correlation between the two?In the fight against covid-19, the skin damage caused by wearing PPE was related to the increased friction between skin and PPE caused by pressure, rapid exercise and sweating.Does consider the increase of friction force under multi-factors when lubricant is used to reduce friction?

3.Experimental products: The author selected four kinds of 19 different products(Table 1 on page7-8) to be used in the same part of the same person in a month. The friction coefficient of each product was measured three times at 1, 2 and 4 hours after use. Will the frequent use of different products change the microclimate of the skin and thus affect the friction coefficient?

Analysis of the results and discussion of this study has the following doubts:

1.Figures 4A to 4F showed schematic rather than measurement results. What does the author want to tell the readers?

2.In the discussion, the authors inferred the results based on these diagrams. Can they reflect the real situation?

3.The authors has come to the following conclusion:Results indicate that the use of emollients and moisturizing creams is to be discouraged when wearing PPE for long duration as they may result in excessive shear forces acting on the skin. Talcum powder, a lanolin containing petrolatum, and a coconut oil-cocoa butter-beeswax mixture provide excellent long-lasting lubrication.But there seems to be little evidence to support.

Minors

1.There are some spelling mistakes in the text.

2.The author described it in the abstract:This research was funded by the Imperial College COVID-19 Response Fund. But on page 10 it was written:The research funder did not play a role in the research.What does that mean?

Reviewer #3: Very interesting and relevant study looking at lubricants to protect skin health in those using PPE during COVID-19. I have some minor points which require clarification.

1. Friction is only one element of the boundary conditions that causes skin damage. Pressure, shear and an altered microclimate will also effect skin health. This is introduced in the manuscript but perhaps a little miss-leading in figure 1. If the same pressure and shear is applied to the skin, its likely to cause similar tissue deformation regardless of lubricant.

2. In addition, Figure 1 should include ‘pressure and shear induced skin injury’ – shear will not be acting in isolation, unless you are referring to tissue shear strain.

3. Did you recreate the material interface of PPE devices when assessing the friction properties? You discuss the implications of skin epidermal absorption, occlusion and water retention, but does this also apply for the PPE interface material?

4. In the introduction it would be worth highlighting PPE devices which are typically implicated in damage, for example respiratory protective equipment (FFP3 masks)

5. NHS England and NHS Improvement [4] do not recommend petrolatum to the skin. This has been reported in one manuscript with limited evidence for its efficacy.

6. Many PPE devices that are implicated in skin damage e.g. the aforementioned FFP3 masks do not employ PDMS at the interface.

https://www.3m.co.uk/3M/en_GB/company-uk/3m-products/~/3M-Aura-Disposable-Healthcare-Respirator-FFP3-Valved-1873V-/?N=5002385+3292799385&rt=rud

7. Brill et al. [17] evaluated the pressures required to secure non-invasive ventilation masks, this may differ to the pressures observed in PPE equipment.

8. I appreciate the forearm is convenient for testing, however does this replicate the skin properties of the face, for example the highly porous sites at the nose? As you’ve highlighted there are large differences in friction for unlubricated skin, which are related to variations in sebum, hydration and mechanical properties

9. Strip waxing the forearm would remove layers of the stratum corneum and result in increased transepidermal water loss. Could this have effected your results? Why not use a commercial hair removal cream?

10. Did you measure the biophysical properties of the skin/check for blanching erythema? Seems like the single participant was put through a large number of mechanical tests.

11. In-between tests the PDMS was applied, so the interactions between skin and PDMS under mechanical loading conditions was not tested? - you have addressed this to some extent in the discussion

12. Results are very descriptive, some inclusion of data, or for example percentage change values from baseline would add context for the reader.

13. No mention of the discussion regarding the functionality of PPE. Indeed, FFP3 masks must retain a seal with the skin in order to filter >95% of particles. Any additional sliding at the interface could be to the detriment of the devices functionality. There must be a careful interplay between maintaining skin health and PPE function.

6. PLOS authors have the option to publish the peer review history of their article (what does this mean?). If published, this will include your full peer review and any attached files.

Reviewer #1: No

Reviewer #2: No

Reviewer #3: **Yes: **Peter Worsley

---

## [Author Response · Author response to Decision Letter 0]

26 Aug 2020

We would like to thank the reviewers for their positive comments and helpful suggestions. Our detailed response to the reviewers comments was uploaded in a separate document.

---

## [Decision Letter · Decision Letter 1]

7 Sep 2020

Evaluating lubricant performance to reduce COVID-19 PPE-related skin injury

PONE-D-20-21415R1

Dear Dr. Masen,

We’re pleased to inform you that your manuscript has been judged scientifically suitable for publication and will be formally accepted for publication once it meets all outstanding technical requirements.

Kind regards,

Jianguo Wang, PhD

Academic Editor

PLOS ONE

Additional Editor Comments (optional):

Reviewers' comments:

Reviewer's Responses to Questions

**Comments to the Author**

1. If the authors have adequately addressed your comments raised in a previous round of review and you feel that this manuscript is now acceptable for publication, you may indicate that here to bypass the “Comments to the Author” section, enter your conflict of interest statement in the “Confidential to Editor” section, and submit your "Accept" recommendation.

Reviewer #1: All comments have been addressed

Reviewer #3: All comments have been addressed

2. Is the manuscript technically sound, and do the data support the conclusions?

Reviewer #1: Yes

Reviewer #3: Yes

3. Has the statistical analysis been performed appropriately and rigorously? 

Reviewer #1: Yes

Reviewer #3: Yes

4. Have the authors made all data underlying the findings in their manuscript fully available?

Reviewer #1: Yes

Reviewer #3: Yes

5. Is the manuscript presented in an intelligible fashion and written in standard English?

Reviewer #1: Yes

Reviewer #3: Yes

6. Review Comments to the Author

Reviewer #1: The revisions are satisfactory to this reviewer. The manuscript is recommended for publication in its current form.

Reviewer #3: Thank you for addressing the comments, I have no further issues. The manuscript has its limitations, although this is in part due to the restrictions in lab based studies during COVID19.

7. PLOS authors have the option to publish the peer review history of their article (what does this mean?). If published, this will include your full peer review and any attached files.

Reviewer #1: No

Reviewer #3: No

---

## [Editor Report · Acceptance letter]

17 Sep 2020

PONE-D-20-21415R1 

Evaluating lubricant performance to reduce COVID-19 PPE-related skin injury 

Dear Dr. Masen:

I'm pleased to inform you that your manuscript has been deemed suitable for publication in PLOS ONE. Congratulations! Your manuscript is now with our production department. 

Kind regards, 

on behalf of

Dr. Jianguo Wang 

Academic Editor

PLOS ONE